



**Quantifying primary and secondary humic-like substances in**
**urban aerosol based on emission source characterization and a**
**source-oriented air quality model**
Xinghua Li[1], Junzan Han[1], Philip K. Hopke[2], Jingnan Hu[3], Qi Shu[1], Qing Chang[1], Qi Ying[4]
[1]School of Space and Environment, Beihang University, Beijing, 100191, China
[2]Center for Air Resources Engineering and Science, Clarkson University, Potsdam, NY USA.
[3]State Environmental Protection Key Laboratory of Vehicle Emission Control and Simulation, Chinese Research
Academy of Environmental Sciences, Beijing 100012, China
[4]Zachry Department of Civil Engineering, Texas A&M University, College Station, TX 77843, USA
*Correspondence to*: Xinghua Li (lixinghua@buaa.edu.cn); Qi Ying (qying@civil.tamu.edu)
**Abstract:** Humic-like substances (HULIS) are a mixture of high molecular weight, water-soluble organic compounds
that are widely distributed in atmospheric aerosol. Their sources are rarely studied quantitatively. Biomass burning is
generally accepted as a major primary source of ambient humic-like substances (HULIS) with additional secondary
material formed in the atmosphere. However, the present study provides direct evidence that residential coal burning is
also a significant source of ambient HULIS, especially in the heating season in northern China based on source
measurements, ambient sampling and analysis, and apportionment with source-oriented CMAQ modeling. Emissions
tests show that residential coal combustion produces 5 to 24% of the emitted organic carbon (OC) as HULIS carbon
(HULISc). Estimation of primary emissions of HULIS in Beijing indicated that residential biofuel and coal burning
contribute about 70% and 25% of annual primary HULIS, respectively. Vehicle exhaust, industry, and power plants
contributions are negligible. Average concentration of ambient HULIS was 7.5 μg/m$^3$ in atmospheric PM$_{2.5}$ in urban
Beijing and HULIS exhibited obvious seasonal variations with the highest concentrations in winter. HULISc account
for 7.2% of PM2.5 mass, 24.5% of OC, and 59.5% of WSOC, respectively. HULIS are found to correlate well with K$^+$,
Cl$^-$, sulfate, and secondary organic aerosol suggesting its sources include biomass burning, coal combustion and
secondary aerosol formation. Source apportionment based on CMAQ modeling shows residential biofuel and coal
burning, secondary formation are important annual sources of ambient HULIS, contributing 57.5%, 12.3%, and 25.8%,
respectively.



## 1 Introduction

Humic-like substances (HULIS) are a mixture of higher molecular weight organic compounds that resemble terrestrial

and aquatic humic and fulvic acids with similar structures and properties (Graber and Rudich, 2006). HULIS are widely

distributed in the atmospheric aerosol, rain, and cloud and fog droplets and account for a significant proportion of the

organic carbon and water-soluble organic carbon (WSOC). For example, Zheng et al. (2013) reported that 9% to 72% of

WSOC is HULIS. Because of their water-soluble and strong surface-active properties, HULIS may play an important

role in the formation of clouds as condensation nuclei, ice nuclei and through aerosol hygroscopic growth (Dinar et al.,

2006; Wang and Knopf, 2011; Gysel et al., 2004). Due to their strong light absorption in the UV range, HULIS can play

an active role as brown carbon in the radiative transfer and photochemical processes (Hoffer et al., 2006). HULIS

deposition can also lead to a decrease in the albedo of ice and snow surfaces (Beine et al., 2011; France et al., 2011;

France et al., 2012). Owing to their redox-active characteristics, HULIS was also suggested to induce adverse health

effect (Lin and Yu, 2011; Ghio et al., 1996; Verma et al., 2012).

In recent years, studies focusing on the spatial and temporal variations, sources, and formation of HULIS have been

reported. A summary of these studies is provided in Table S1. Previous studies have identified primary emission and

atmospheric secondary formation as the important sources of HULIS. Among the primary emission sources, biomass

burning is generally accepted as a major source of HULIS, with the evidence from ambient and source sampling as well

as source apportionment studies (Lin et al., 2010a, b; Kuang et al., 2015; Park and Yu, 2016; Schmidl et al., 2008a, b;

Goncalves et al., 2010). Recently, residential coal burning was suggested as an important primary HULIS source during

winter (Tan et al., 2016; Voliotis et al., 2017). However, direct evidence of HULIS emission from coal combustion is

limited. Only one study on HULIS emitted from residential coal combustion was reported and the results showed that

HULIS accounted for 5.3% of smoke $PM_{2.5}$ (Fan et al., 2016). Unfortunately, only a light coal in the shape of

honeycomb briquette was tested that did not reflect the variety of coal types and forms actually being used for

residential heating and cooking in China. Another possible primary HULIS source is vehicle exhaust although there is

uncertainty in the importance of this source (El Haddad et al., 2009; Salma et al., 2007; Lin et al., 2010b; Kuang et al.,

2015). No direct evidence of primary HULIS in vehicle exhaust has been reported. Secondary processes, including

formation in the cloud droplets, heterogeneous or aerosol–phase reactions, can be important sources of HULIS (Lin et

al., 2010b; Zheng et al., 2013).

Previous studies of HULIS source identification were generally based on the relationship between HULIS and the

tracers for specific sources (such as K, levoglucosan, Cl⁻, etc.) (Voliotis et al., 2017; Tan et al., 2016; Lin et al, 2010;





Park and Son, 2016; Baduel et al., 2010). Those correlation analyses between HULIS and some species may provide
some information regarding possible source and formation of HULIS. However, they do not provide quantitative source
apportionments. To date, studies that quantitatively identify HULIS sources are limited (Kuang et al., 2015; Srivastava
et al., 2018). Kuang et al. (2015) applied positive matrix factorization (PMF) to apportion sources of ambient HULIS in
the Pearl River Delta (PRD) in Southern China. Their study showed that secondary formation was the most important
source of HULIS throughout the year with an annual average contribution of 69% at an urban site. Biomass burning
also contributed significantly to ambient HULIS.
Thus, information is scarce on the quantitative apportionment of HULIS sources in the more polluted regions in
Northern China, especially in the winter when large quantities of coal are consumed for indoor heating. Moreover, a
considerable proportion of coal is burned in residential household stoves in rural, suburban and even some urban areas
under poor combustion conditions and without any emission controls. This coal combustion results in high air pollutant
emissions and high ambient concentrations. Wang et al. (2016) estimated that more than 30 million tons of coal are
burned per year in households in just the Beijing, Tianjin, and Hebei (BTH) region in Northern China. Residential
sources in the BTH region contributed to 32% and 50% of primary $PM_{2.5}$ emissions over the whole year and in winter,
respectively (Liu et al., 2016).
The primary goals of this study are to determine whether residential coal combustion is a significant source of ambient
HULIS and quantify its contributions to HULIS in Beijing. Given the large vehicle population in Beijing (up to 5.2
million in 2012), this study also provides a chance to examine the vehicular emissions contribution to ambient HULIS.
Studies included: (1) Characterization of the HULIS emitted from residential coal stoves, vehicle exhaust, and
residential biofuel burning; (2) Estimation of anthropogenic primary emission of HULIS based on these source
measurements; (3) Measurement of HULIS concentrations and other major species in the ambient urban Beijing $PM_{2.5}$
from June 2012 to April 2013; and (4) Application of the source-oriented Community Multiscale Air Quality (CMAQ)
model to quantitatively determine the source contributions to HULIS. The information obtained in this study improves
our understanding of the characteristics and sources of primary HULIS and the impact of those sources on HULIS in
ambient $PM_{2.5}$.
**2 Materials and Methods**
**2.1 Ambient sampling**
Beijing is surrounded by mountains to the west, north, and northeast, and is adjacent to the northwest portion of the
North China Plain. It has a warm and semi-humid continental monsoon climate with four distinctive seasons,



characterized by strong windy and dusty weather in spring, high temperatures and humidity with extensive rain in
summer, cool and pleasant weather in autumn, and cold and dry weather in winter. The annual average wind speed is
2.5 m s$^{-1}$ with mostly northerly or northwesterly winds in winter and southerly or southeasterly winds in summer.
Twenty-four-hour ambient PM$_{2.5}$ samples were collected non-continuously from 14 June 2012 to 2 April 2013 on the
campus of Beihang University (BHU, 39°59'N, 116°21'E) (Figure S1). The sampling period covered four seasons with
132 samples being collected for HULIS analysis. The site is surrounded by educational and residential districts without
major industrial sources. Major nearby roads are the North Fourth Ring Road about 900 m to the north, North Third
Ring Road about 1.2 km to the south, and Xueyuan Road about 350 m to the east. Ambient PM$_{2.5}$ sampling instruments
were installed on the roof of a building approximately 25 m above the ground level at Beihang University. A
high-volume aerosol sampler (RFPS-1287-063, Thermo, USA) was operated at a flow rate of 1.13 m$^3$ min$^{-1}$ to collect
PM$_{2.5}$ samples on prebaked quartz filters (with area 417.6 cm$^2$) for the determination of water-soluble organic carbon
(WSOC) and humic-like substances (HULIS). PM$_{2.5}$ samples were also collected using a 5-channel Spiral Ambient
Speciation Sampler (SASS, Met One Inc., USA) with a flow rate of 6.7 L min$^{-1}$. Wang et al. (2015) provided the details
of the sample collection.
Meteorological data including wind speed (WS), temperature, relative humidity (RH) and precipitation were obtained
from China Meteorological Data Sharing Service System (http://cdc.cma.gov.cn/home.do).
**2.2 Source Sampling**
Residential biofuel and coal combustion emissions, and vehicle exhaust, which are representative of typical emission
sources around Beijing, were sampled in this study.
**2.2.1 Residential biofuel and coal combustion**
Three typical types of biofuel, i.e. wheat straw, corn stover, and wood, were burned in an improved stove, which has an
enclosed combustion chamber and a bottom grate and a chimney. The sampling procedures are described by Li et al.
(2007, 2009) and are briefly summarized here. The water boiling test was used to simulate a common cooking
procedure. The burning cycle included heating a specific amount of water from room temperature to its boiling point
and then letting it simmer for a few minutes. Both the high power and low power phases were included in the burn
cycle to simulate cooking practices of a typical household. The sampling period covered the entire cycle and lasted
20-30 minutes.
Five coal types were selected for source testing covering a wide range of maturity with volatile matter content varying
from 9.6% to 32.4%. Two coal stoves were tested, including a high efficiency, heating stove that employs under-fire



combustion technology and a traditional cooking and heating stove that employs over-fire combustion technology (Li et
al., 2016). Four chunk coals and one briquette coal were burned in the high efficiency stove and three chunk coals were
burned in the traditional stove. Coal/stove combinations are presented in Table 2. To reduce the interference from
igniting the fire, coal was ignited with a propane gas flame from a torch. Emission sampling covered from fire start to
fire extinction and lasted two to three hours.
Source testing of residential biofuel and coal combustion was performed at Beihang University. The test fuels were
air-dried, and the results of their proximate and ultimate analyses are listed in Table S1 in SI. An outline of the sampling
system is shown in Fig. S2. The stove was placed into a chamber. Purified air was induced into the chamber with a fan
to provide dilution air. Emissions were extracted from the chimney with an exhaust hood and were diluted with purified
air, cooled to no more than 5 degrees Celsius at ambient temperature, and then drawn through a sampling duct and
exhausted from the laboratory. Both air flows were adjusted using frequency modulators to change fan speeds. The gas
flow velocity in the sampling duct was measured by a pitot tube to be over 5 m/s. Flow was isokinetically withdrawn
from the sampling duct with a probe and directed into the residence chamber. $PM_{2.5}$ samples were collected from the
end of the residence chamber onto prebaked quartz filters with a diameter of 47mm through $PM_{2.5}$ cyclones at a flow
rate of 16.7 liters/min.
**2.2.2 Vehicle exhaust**
Four light-duty gasoline vehicles certified as meeting the China 4 emissions regulations were tested for their emissions
on a chassis dynamometer. The tests were conducted using the New European Driving Cycle (Marotta, et al., 2015) and
lasted 1180 s, including four repeated urban driving cycles and one extra-urban driving cycle. The emissions testing and
sampling system are described in detail by Li et al. (2016) and are briefly summarized here. The vehicle exhaust was
directed into a critical flow Venturi constant volume sampler in a full flow dilution tunnel. The $PM_{2.5}$ samples were
collected on prebaked quartz filters with a diameter of 47mm through $PM_{2.5}$ cyclones at a flow rate of 80 L/min.
Three heavy-duty diesel trucks were selected to perform on-road emission test. The tests were conducted on both
highway and city roads. The emission testing and sampling system are described in detail elsewhere (He et al., 2015)
and are briefly summarized here. A Micro Proportional Sampling System (SEMTECH-MPS; Sensors Inc., MI, USA)
was used to draw a constant ratio of sample flow from exhaust and dilute the sample flow. $PM_{2.5}$ samples were collected
onto prebaked quartz filters with a diameter of 47mm through $PM_{2.5}$ cyclones at a flow rate of 10 liters/min.
Tunnel measurements was also conducted to collect vehicle exhaust at the Badaling Tunnel in Beijing. The length of the
tunnel is 1085 m. The ventilation in the tunnel was achieved by the flow of the traffic induced into the tunnel during the
sampling period. $PM_{2.5}$ samplers with prebaked 47mm quartz filters were operated at a flow rate of 16.7 L/min at the




inlet and the outlet of the tunnel simultaneously. The sampling period was 2 hours and the samples represent the mixed
exhaust of gasoline-fueled vehicles and diesel-fueled vehicles.
All source samples collected on the quartz filters were analyzed for HULIS, WSOC and OC/EC according the methods
described in Section 2.3.
**2.3 Chemical Characterization**
HULIS isolation was based on the extraction method developed by Varga et al. (2001) and used in many other studies
(Nguyen et al., 2014; Lin et al., 2010b; Fan et al., 2012; Song et al., 2012; Lin et al., 2011; Salma et al., 2013; Feczko et
al., 2007; Krivácsy et al., 2008). The separation procedure is provided in SI Text S1. WSOC and $HULIS_C$ were
determined using a total organic carbon (TOC) analyzer (Shimadzu TOC-Vcph, Japan) based on a
combustion-oxidation, non-dispersive infrared absorption method. The TOC was determination by subtracting inorganic
carbonate (IC) from total carbon (TC): TOC = TC - IC. The reported data were the average results of three replicate
measurements. Mass concentrations of HULIS were obtained from $HULIS_C$ by multiplying a scaling factor of 1.9 as
suggested by Lin et al. (2012a), Kiss et al. (2002), and Zheng et al. (2013).
A 0.5 $cm^2$ punch from each quartz filter was analyzed for OC and EC using a DRI Model 2001 Thermal/Optical Carbon
Analyzer (Atmoslytic Inc., Calabasas, USA) following the IMPROVE-A thermal optical reflectance (TOR) protocol
(Chow et al., 2007).
The $PM_{2.5}$ samples from SASS were also analyzed for mass, water-soluble inorganic ions analysis as described by
Wang et al. (2015).
**2.4 CMAQ modelling of primary HULISc**
A source-oriented version of the Community Multiscale Air Quality (CMAQ) model (version 5.0.1) was used in this
study to track primary $PM_{2.5}$ ($PPM_{2.5}$) from different emission sectors and determine the resulting concentrations of
primary HULIS. The model was used in a previous study to determine source contributions to $PPM_{2.5}$ mass, EC and
primary OC (POC) in China. Details of the source apportionment technique can be found in Hu et al (2015). In
summary, source contributions to $PPM_{2.5}$ mass were directly determined using non-reactive source-specific tracers to
track the emissions of $PPM_{2.5}$ from different sources. These non-reactive tracers were treated identically to the other
PPM components when simulating their emission, transport, and removal. A constant scaling factor (typically $10^{-4}$ or
$10^{-5}$) was used to scale the actual emission rate of these tracers to ensure that their concentrations are sufficiently low
that they do not alter the removal rates of other PM components. The concentrations and source contributions to EC and
POC were determined during post-processing by using source-specific emission factors as well as the model predicted





source contributions to PPM$_{2.5}$ mass concentrations. This technique can be used to determine source contributions to
primary HULIS. For example, contributions of the i$^{th}$ emission source to primary HULISc concentration (HULISc$_{,i}$) can
be calculated using equation (1):
$HULISc_{,i} = PPM_{2.5,i} * f_{OC,i} * f_{HULIS,i}$        (1)
where f$_{HULIS,i}$ is the mass fraction of HULIS per unit emission of POC from the i$^{th}$ source (see Section 3.3 below for
estimation of HULIS primary emission) and f$_{OC,i}$ is the mass fraction of POC per unit emission of PPM$_{2.5,i}$ from the i$^{th}$
source, and PPM$_{2.5,i}$ is the calculated source contributions to PPM$_{2.5}$ from the i$^{th}$ source based on the non-reactive tracer.
The total concentration of primary HULIS can be determined by adding the primary HULIS contributions from the
different sources.
In this study, the model uses a 36 km × 36 km horizontal resolution to cover a rectangular domain that includes all of
China and neighboring countries. Source contributions to HULIS were calculated for the periods when observations of
HULIS are available. Emissions from anthropogenic source sectors (residential sources, power plants, industries, and
transportation) are based on Multi-resolution Emission Inventory of China (MEIC) 2012 (www.meicmodel.org). Open
biomass burning was estimated using the FINN dataset from the National Center for Atmospheric Research (NCAR)
(Wiedinmyer et al., 2011). Natural emissions from soil erosion and sea spray were modeled within the CMAQ model
(Appel et al., 2013; Kelly et al., 2010). Biogenic emissions were estimated using the Model for Emissions of Gases and
Aerosol from Nature (MEGAN) version 2.10. Meteorological fields were calculated using the Weather Research and
Forecasting (WRF) model. Details of the model setup, input data preparation, as well as model evaluation for PPM$_{2.5}$
mass are documented by Hu et al (2015). In this study, a comparison of predicted daily PPM$_{2.5}$ concentrations with
observations was performed and only those predictions with fractional errors (FE) less than 0.6 were included in the
calculation of primary HULIS. The values of f$_{OC}$ for different source sectors used in the calculation are included in SI
Table S3. These values were used in Ying et al. (2018), and the predicted daily-average POC and EC concentrations
generally agree with predictions for both daily and annual average concentrations.
**3 Results and discussion**
**3.1 General of ambient aerosol characteristics**
The concentrations of PM$_{2.5}$, carbonaceous species (OC, EC, WSOC and HULIS), and inorganic ions such as SO$_4^{2-}$,
NO$_3^-$, NH$_4^+$, and K$^+$ are summarized in Table 1. The 24-hour average PM$_{2.5}$ concentration for the sample set was 106 ±
89 μg/m$^3$, about three times the national annual air quality standard (35 μg/m$^3$). The highest concentration (~600 μg/m$^3$)
appeared on 12-13 January 2013 as reported in other studies (Quan et al., 2014; Tian et al., 2014; Zheng et al., 2015).





The severe pollution events were always accompanied by high relative humidity and low wind speeds (Fig. 1). During
the entire sampling period, severely polluted days (PM$_{2.5}$ concentrations ≥ 150 μg/m$^3$) constituted about 22%, while fair
days (PM$_{2.5}$ concentrations ≤ 75 μg/m$^3$) approached 50%. The average PM$_{2.5}$ concentrations in summer, autumn, winter,
and spring were 98 ± 60 μg/m$^3$, 58 ± 48 μg/m$^3$, 150 ± 121 μg/m$^3$, and 120 ± 76 μg/m$^3$, respectively.
The average HULIS concentration for the study period was 7.5 ± 7.8 μg/m$^3$.   This value is lower than the average
value of 11.8 μg/m$^3$ measured at a rural site in the PRD region that was heavily influenced by biomass burning (Lin et
al., 2010b). However, it is higher measurements in the urban areas (about 5 μg/m$^3$) in the PRD (Lin et al., 2010a; Kuang
et al., 2015), urban Shanghai (about 4 μg/m$^3$) (Zhao et al., 2015), and urban Lanzhou (about 4.7 μg/m$^3$) (Tan et al.,
2016). HULIS exhibited obvious seasonal variations as shown in Figure 1 and Table 1. The seasonal average
concentrations were 5.5 ± 4.4 μg/m$^3$, 5.6 ± 4.7 μg/m$^3$, 12.3 ± 11.7 μg/m$^3$, and 6.5 ± 5.5 μg/m$^3$ in summer, autumn,
winter, and spring, respectively. The winter mean was about twice the value in any other season, and the highest
concentration (54.96 μg/m$^3$) of HULIS was observed on the same day that the highest concentration of PM$_{2.5}$ was
observed. The mean HULIS concentrations were very similar between summer and autumn in contrast with PM$_{2.5}$ that
has much higher concentrations in the summer (Table 1). These seasonal variations were similar with those observed in
Aveiro and K-puszta (Feckzo et al., 2007), but those annual average concentrations (about 2.4 μg/m$^3$ and 3.2 μg/m$^3$,
respectively) were much lower than in Beijing. The concentrations of HULIS in previously reported studies are
summarized in Supporting Table S1.
HULIS and PM$_{2.5}$ had a significant positive correlation with the annual r$^2$=0.90 (r$^2$ = 0.77, 0.96, 0.96 and 0.94 in
summer, autumn, winter, and spring, respectively) (Figure S4a). The seasonal average of HULIS/PM$_{2.5}$ was 5.9%, 9.4%,
7.9%, and 4.8% in summer, autumn, winter, and spring, respectively. The annual average of HULIS/PM$_{2.5}$ was 7.2% ±
3.3%, lower than the ~10% in the PRD region (Lin et al., 2010a).
Strong correlations of HULIS$_C$ with OC were observed with the annual r$^2$=0.87 (r$^2$ = 0.94, 0.82, 0.89 and 0.84 in
summer, autumn, winter, and spring, respectively) (Fig S4c). The percentage of HULIS$_C$ in OC for summer, autumn,
winter, and spring, respectively, were 29.2 ± 6.2%, 26.2 ± 9.6%, 21.0 ± 7.1%, and 22.0 ± 6.9% with an annual average
of 24.5% ± 8.3%.
Strong correlations of HULIS$_C$ with WSOC were also observed with the annual r$^2$=0.98 (r$^2$ = 0.99, 0.96, 0.99 and 0.98
in summer, autumn, winter, and spring, respectively) (Figure S4b). The percentage of HULIS$_C$ in WSOC for summer,
autumn, winter, and spring, respectively, were 66.7 ± 5.4%, 54.1% ± 11.2%, 62.3% ± 5.7% and 56.6% ± 6.3% with an
annual average of 59.5% ± 9.2%, suggesting that HULIS$_C$ was the major constituent of WSOC. This value is
comparable to the results (about 60%) at urban sites in the PRD region (Lin et al., 2010b; Fan et al., 2012), Shanghai



(Zhao et al., 2015), Korea (Park et al., 2012), Budapest (Salma et al., 2007; 2008; 2010), and high-alpine area of the
Jungfraujoch, Switzerland (Krivácsy et al., 2001). However, it is higher than the rural areas in K-puszta, Hungary
(Salma et al., 2010) and the northeastern US (Pavlovic and Hopke, 2012). The ratios of HULIS$_C$/WSOC reported by
previous studies are listed in Supporting Table S1.

**3.2 HULIS emission characteristics from various sources**

The measured HULIS$_C$/OC (i.e. f$_{HULIS,i}$), HULIS$_C$/WSOC from the source samples are presented in Table 2. Biomass
combustion produces a significant fraction of HULIC in OC (0.41-0.50) whether burning wood or crop straw. Those
values are high compared to previous studies. The HULIS$_C$/OC values obtained by Lin et al., (2010a, 2010b) were 0.14
to 0.34 from rice straw and sugarcane burning in the PRD region in south China. Park and Yu (2016) found the ratios
from burning rice straw, pine needles, and sesame stems in Korea were in the range of 0.15 to 0.29. Schmidl et al.
(2018a, 2018b) reported the ratios of 0.01-0.12 for wood burning and 0.33-0.35 for leaves burning in the mid-European
Alpine region. Goncalves et al. (2010) obtained ratios of 0.04 to 0.11 from wood burning in Portugal. HULIS is an
important component of WSOM. High HULIS$_C$/WSOC ratios (0.62 to 0.65) were observed for three types of biomass
burning in this study. These results are comparable with two previous studies. Fan et al. (2017) reported the ratios from
burning rice straw, corn straw, and pine branch were in the range of 0.57 to 0.66. Park and Yu (2016) obtained ratios in
the range of 0.36 to 0.63 from burning three types of biomass. However, Lin et al. (2010a) reported relatively low
values ranging from 0.30 to 0.33 from rice straw and sugarcane burning.
Residential coal combustion produces 5 to 24% of the OC as HULIS for all the coal/stove combinations in this study.
Only one prior study measured HULIS emitted from residential honeycomb coal briquette combustion (Fan et al., 2016).
However, the HULIS to OC ratio was not reported in that study. HULIS/WSOM ratio (0.46) in that study are
comparable with our HULIS$_C$/WSOC data (0.41-0.62).
Light-duty gasoline and heavy-duty diesel vehicles also produced primary HULIS on the order of 5 to 16% of the
emitted OC. The HULIS content detected in the vehicle exhaust samples was generally less than the detection limit for
these measurements. Thus, these reported ratios of HULIS$_C$ to OC for vehicle emissions have high uncertainties. Ratios
of HULIS$_C$ to OC for vehicle emissions obtained in this study are much higher than prior tunnel measurements (2-5%)
(El Haddad et al. 2009). However, they are comparable with those from residential coal combustion. Due to more
complete combustion or more advance emission controls in vehicles than residential solid fuel combustion, OC
emission factors for vehicles are normally around two orders of magnitude less than that for residential coal combustion
(MEP of China, 2014), so HULIS emission from vehicles can be neglected as described in Section 3.3.





**3.3 Estimation of HULIS primary emission**

The average values of $f_{HULIS,i}$ for residential biofuel and coal combustion, and vehicle exhaust obtained from our measurement were used for to assess the extent of primary emissions. Due to lack of $f_{HULIS,i}$ for the other sectors, such as power plants and industries, considering combustion/production technology and emission control technology, we assumed values for these two sectors as 0.01 and 0.05, respectively.

Based on OC emissions for different sources in the MEIC inventory and the $f_{HULIS,i}$ for the various sources described above, the annual anthropogenic primary emission of HULIS in Beijing is estimated to be approximately 6.3 Gg with over 60 percent of this primary HULIS being emitted during the heating season. Residential biomass and coal burning contribute about 70% and 25% of the annual primary HULIS emissions, respectively. Vehicle exhaust contributions to annual primary HULIS emission are negligible (less than 2%). While industry sector and power plants contribute about 3% and close to zero, respectively. In winter, residential biomass and coal burning contribute close to 98 percent of primary HULIS (Supporting Table S3).

Terrestrial and marine emissions were not included in these estimations of primary HULIS emissions since they were considered to be negligible for inland cities, such as Beijing (Graber and Rudich, 2006; Zheng et al., 2013).

**3.4 Possible primary sources and secondary formation of HULIS**

Ambient HULIS sources include primary sources and atmospheric secondary processes that convert gaseous precursors to HULIS. The correlation between HULIS and other measured constituents provide information regarding possible sources and formation mechanisms of HULIS.





### 3.4.1 HULIS from primary sources

Correlations between HULIS and primary species in $PM_{2.5}$ are shown in Figure 2. POC and secondary organic carbon

(SOC) were estimated using the EC tracer method (Lim and Turpin, 2002; Turpin and Huntzicker, 1995). The details of

the method and evaluation are provided in Text S2. Figures 2a and 2b show that there are strong correlations between

HULIS and POC, and HULIS and EC throughout the year indicating that HULIS has sources and/or transport processes

similar to those of POC and EC. Both POC and EC are co-emitted by the incomplete combustion of carbon-containing

fuels. According to the 2010 MEIC data for Beijing 2010, biomass and residential coal burning contribute more than 80

percent of the POC emissions, the industrial sector contributes over 10 percent, and vehicular exhaust contributions are

negligible. For EC emission, residential coal burning contributes more than 50 percent, biomass burning, industry, and

vehicles contributes the rest.

$K^+$ generally originate from biomass burning with lesser contributions from coal burning and dust. However, biomass

burning is regarded as the most important source for $K^+$ and it is often used as an indicator of biomass burning (Kuang

et al., 2016; Zhang et al., 2013; Park et al., 2015; Pio et al., 2008; Wang et al., 2011; 2012; Cheng et al., 2013). In North

China, biomass burning occurred in all seasons including residential cooking, heating, and open biomass burning

(Cheng et al., 2013; Zheng et al., 2015). High $K^+$ concentrations in this study were observed with mean values of 2.2 ±

2.9 µg/m³, 1.3 ± 1.0 µg/m³, 3.2 ± 3.6 µg/m³ and 2.2 ± 1.3 µg/m³ in summer, autumn, winter, and spring, respectively,

and an annual average of 2.2 ± 2.6 µg/m³. As shown in Figure 3c, HULIS and $K^+$ exhibited a strong correlation with

$r^2$=0.76, 0.73, and 0.61 in summer, autumn, and spring, respectively, suggesting the contribution of biomass burning to

HULIS. During the winter sampling period, a low correlation was initially obtained ($r^2$ = 0.21). However, two extreme

values of $K^+$ were observed on New Year's Eve (February 9, 2013, 14.6 µg/m³) and Lantern Festival (February 24, 2013,

17.6 µg/m³). Prior studies had suggested that fireworks during the Spring Festival and Lantern Festival produce very

high $K^+$ concentrations (Shen et al., 2009; Jing et al., 2014; Jiang et al., 2015). Excluding these two days (red points in

Figure 2c), the correlation between HULIS and $K^+$ increased to $r^2$=0.73, indicating the contribution of biomass burning

to HULIS in winter. The strong correlation coefficient between HULIS and $K^+$ across all the seasons also confirmed that

biomass burning was a significant primary HULIS emission source as presented in the Section 3.3.

$Cl^-$ is usually believed to be associated with coal combustion and biomass burning (Yu et al., 2013; Gao et al., 2015;

Yao et al., 2002; Li et al., 2007; Li et al., 2009). A significant contribution from sea-salt particles for $Cl^-$ in $PM_{2.5}$ can be

excluded since the average mole ratios of $Cl^-$ to $Na^+$ across four seasons in this study is 5.0, much higher than the ratio

in seawater of 1.17. Moreover, the sampling site in Beijing is about 200 Km from the sea. The correlation of HULIS

and $Cl^-$ is shown in Fig. 2d. In winter and spring, HULIS is moderately correlated with $Cl^-$ with $r^2$=0.56 and $r^2$=0.64,



respectively. While weaker correlations were observed in summer and autumn with r$^2$=0.40 and r$^2$=0.43, respectively.
This result reflects the different amounts of coal burned in specific seasons. In winter and spring in northern China, coal
combustion for heating was quite prevalent and more coal was burned compared to the other two seasons, resulting in
the substantial emissions of gaseous and particulate pollutants, including HULIS and Cl$^-$. The source study in Section
3.2 found that HULIS contributed to about 12% of OC emitted from residential coal combustion. The correlation
coefficient between HULIS and Cl$^-$ in winter and spring provides additional support for coal burning being an important
primary HULIS emission source as discussed in Section 3.3. The strong correlation between HULIS and Cl$^-$ in winter
(R$^2$=0.89) and weak correlation in summer (R$^2$=0.17) were also revealed in Lanzhou, another city in northern China
(Tan et al., 2016). Significant correlation between HULIS and Cl$^-$ in wintertime urban aerosols from central and
southern Europe were also found (Voliotis et al., 2017). The authors suggest the high concentration of HULIS during
winter was probably related with residential coal burning (Tan et al., 2016; Voliotis et al., 2017).
Ca$^{2+}$ would be more likely originated from the re-suspended road dust and long-range transported dust (Gao et al.,
2014). The poor correlation between HULIS and Ca$^{2+}$ (as shown in Figure 2e) indicated dust was not likely to be an
important source of HULIS.
**3.4.2 HULIS associated with atmospheric secondary processes**
The correlations between HULIS and related secondary species are shown in Figure 3. As shown in Figure 3a and 3b,
HULIS correlated well with SO$_4^{2-}$ and SOC (R$^2$=0.68 for HULIS and SO$_4^{2-}$ and R$^2$=0.61 for HULIS and SOC),
suggesting that HULIS and secondary species may have similar formation pathways. Moderate to strong correlations of
HULIS and SO$_4^{2-}$, and HULIS and SOC were observed in autumn and winter, but with significant differences in the
concentrations. The lower temperature and solar intensity in winter were not conducive to the photochemical formation
of secondary aerosols, but high relative humidity and stable synoptic meteorological conditions accompanied with
heterogeneous reactions probably played a role in the formation of secondary aerosols (Zheng et al., 2015). In summer,
HULIS was strongly correlated with SOC (R$^2$=0.85), while weakly correlated with SO$_4^{2-}$ (R$^2$=0.41), indicating the
distinct formation processes of HULIS and SO$_4^{2-}$. High temperature and solar radiation accelerated the photochemical
reactions between oxidants and organic precursors (Lin et al., 2010). However, a moderate correlation of HULIS &
SO$_4^{2-}$ (R$^2$=0.36) but unfavorable correlation between HULIS and SOC (R$^2$=0.10) were acquired in spring, suggesting
the different formation pathways of HULIS and SOA.
**3.5 HULIS source apportionment based on CMAQ modelling**
CMAQ predicted concentrations of PPM$_{2.5}$ from different sources were used to calculate HULISc from these sources





using equation (1). The total concentration of primary HULIS can be determined by adding up primary HULIS from
different sources. Figure 4 shows the predicted primary HULISc and observed HULISc concentrations with the
prediction uncertainty. Only days with acceptable $PPM_{2.5}$ performance were shown in the Figure 4. Primary HULISc in
January and March 2013 accounts for almost all observed HULISc in these two months. In summer and autumn 2012,
predicted primary HULISc concentrations are approximately 1-2 µg m$^{-3}$. There were days when the observed HULISc
concentrations were much higher than predicted primary HULISc concentrations indicating potential contributions of
secondary HULISc.
Table 3 shows the seasonal contributions for each source as well as average source contributions for the whole sampling
period to ambient HULIS in Beijing based on the observed total HULISc and CMAQ predicted primary HULISc on the
days with acceptable $PPM_{2.5}$ performance. Contributions of HULISc from secondary processes were determined by
subtracting predicted primary HULISc from observed HULISc. For those days when the predicted primary HULISc
concentrations are greater than the observed HULISc, the predicted primary HULISc concentrations were set to equal
the observed HULISc and the secondary HULISc were set to zero. Based on the HULIS emissions from residential
biofuel and coal burning described in Section 3.3, the contributions of biofuel and coal burning in the residential sector
to ambient HULIS were estimated separately.
Overall, residential biofuel burning was the most important source of ambient HULIS, contributing more than half of
the ambient HULIS concentrations, much higher than those results from the PRD in Southern China (less than 20%)
(Kuang et al. 2015). This difference is likely with the result of greater biofuel burning during the heating seasons in the
Beijing area. Residential coal burning contributes 12.3±2.8% to ambient HULIS and is also a significant source of
ambient HULIS. A large contribution from residential sector to ambient HULIS is consistent with the estimation of
HULIS primary emission and the correlations between HULIS and primary species previously presented. Vehicle
emissions and other primary sources, such as industries, contribute negligible amounts to the ambient HULIS.
Contributions from the residential sector display strong seasonal variations. In winter and spring, residential biofuel and
coal burning accounted for over 80% of the total HULISc while their contributions were reduced to approximately 40%
in summer and autumn. The seasonal variations were a reflection of seasonal pattern of those activities in this region.
Secondary formation is estimated to have contributed an average of 25.8±9.3% to the HULIS concentrations and was
another major source to ambient HULIS as indicated by the correlations between HULIS and secondary species (i.e.
SNA, SOC). However, our result is much lower than those results from PRD in Southern China (55 to 69%) (Kuang et
al. 2015). The difference is driven by the differences in sources and climatological patterns between these two sites.
There is much greater combustion for space heating in the colder north and atmospheric reaction rates will be higher in





the warmer south. Contributions from secondary processes also show obvious seasonal variations trend. In winter and
spring, secondary processes accounted for less than 20% of the total HULISc with large uncertainties while their
contributions were increased to 40±18% and 53±17% in summer and autumn. Higher secondary contributions were also
found during warm seasons in the PRD region (Kuang et al. 2015). In addition to the proposed heterogeneous
secondary formation pathways for HULISc, oxidation reactions initiated by chlorine (Cl) radicals can form SOA (Wang
and Ruiz, 2017; Riva et al., 2015). Thus, Cl release by coal combustion may have the potential to contribute to HULISc,
especially during winter when OH radical concentrations are much lower (monthly average $5.5\times10^{-3}$ ppt for winter vs.
$1.25\times10^{-1}$ ppt for summer based on CMAQ calculations for Beijing). However, the concentrations of secondary
HULISc for winter estimated in this study are uncertain ($1.8 \pm 2.2$ µg m$^{-3}$) compared to the summer time average
concentration ($1.0 \pm 0.4$ µg m$^{-3}$). Therefore, the role of Cl initiated reactions producing HULISc cannot be definitively
determined.
Figure 5 shows scatter plot of predicted primary HULISc vs observed HULISc concentrations. Moderate to strong
correlations between predicted primary HULISc were observed in winter and spring, while relatively weaker correlations
were found in autumn. Moreover, low correlations were observed in summer. The variation of correlation coefficient
between predicted primary HULISc and observed HULISc in different seasons also provides additional support for the
relative importance of primary and secondary HULIS as shown in Table 3.
**Supporting Information**
The supporting information file for this paper provides the details of HULIS analytical procedures, and prior literature
regarding HULIS in the ambient aerosol. It also provides some additional figures and descriptions that help to support
the analyses and discussion presented in the paper.
**Acknowledgment**
This work was supported by the National Nature Science Foundation of China (Grant No. 41575119, 41275121) and the
National Key Research and Development Program of China (No. 2017YFC0211404). The authors also want to
acknowledge the Texas A&M Supercomputing Facility (http://sc.tamu.edu) for providing computing resources useful in
conducting the CMAQ simulations reported in this paper.

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





**Figures**

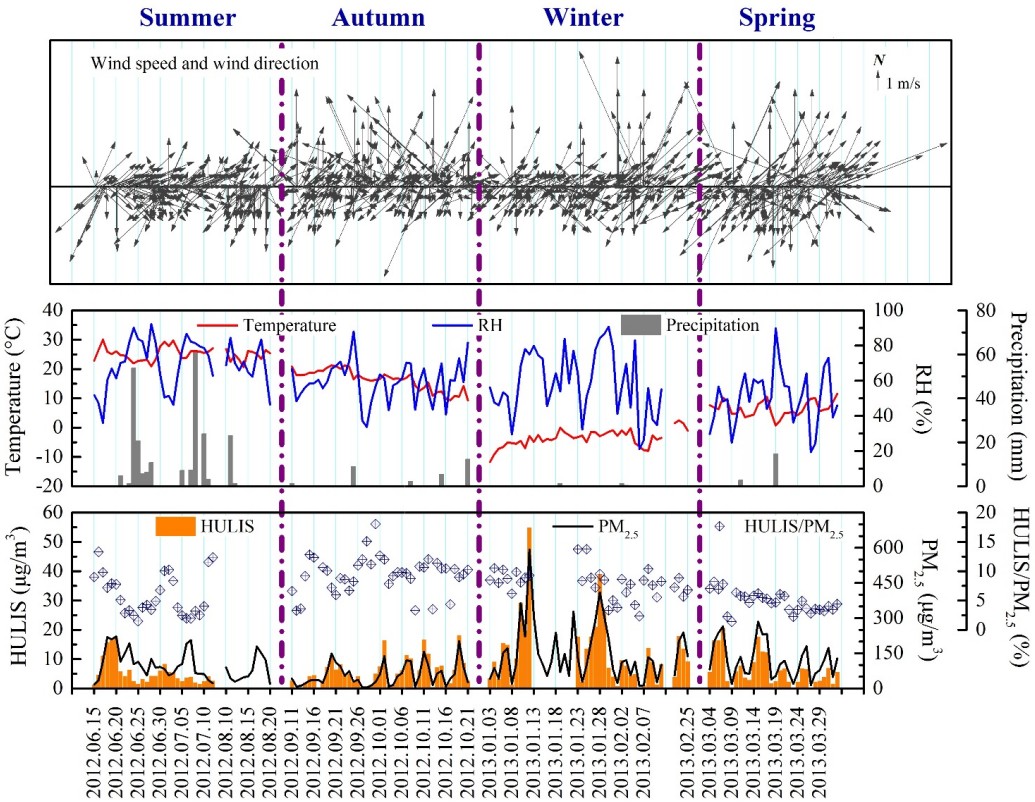


**Figure 1.** Time series of meteorological data (wind speed, wind direction, temperature, relative humidity and
precipitation), HULIS, PM$_{2.5}$ and HULIS/PM$_{2.5}$ for the sampling period.


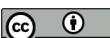







**Figure 2.** Correlations between HULIS and POC (a), HULIS & EC (b), HULIS & K$^+$(c), HULIS & Cl$^-$(d), HULIS &

Ca$^{2+}$(e). Concentrations in four seasons are represented by different shaped points with different colors. Linear

regressions are also given with corresponding equations.






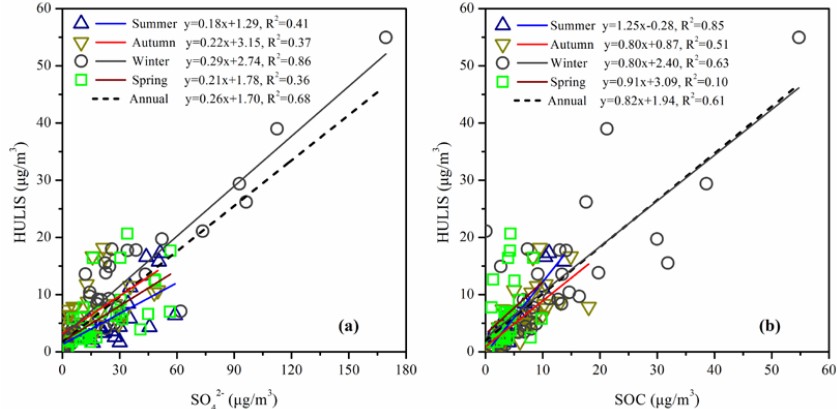


**Figure 3.** Correlations between HULIS & $SO_4^{2-}$ (a), HULIS and SOC (b). Concentrations in four seasons are

represented by different shaped points with different colors. Linear regressions are also given with corresponding

equations.







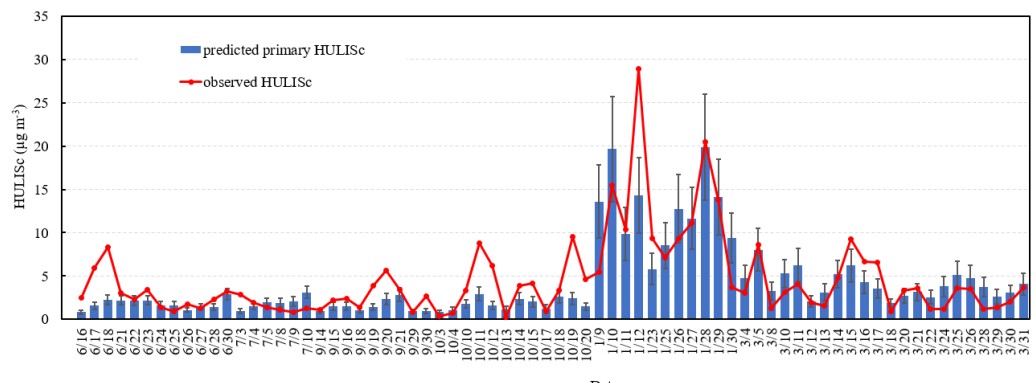


**Figure 4.** Predicted primary HULISc and observed HULISc concentrations on the days with relatively good primary

PM$_{2.5}$ model performance. Error bar is the standard deviation of prediction, which is calculated as described in SI Text
S3.1.






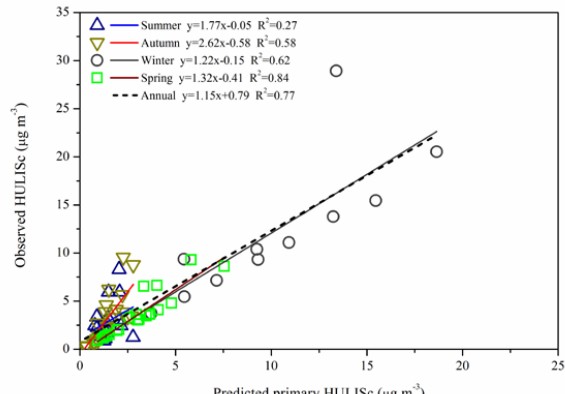


**Figure 5.** Scatter plot of predicted primary HULISc and observed HULISc concentrations. Concentrations of each
seasons are represented by different shaped points with different colors. Linear regressions are also given with
corresponding equations.





**Tables**
**Table 1.** Summary of the concentrations of $PM_{2.5}$, carbon species, water-soluble ions and percentages of several species
to some others.

| Species | Average | Summer | Autumn | Winter | Spring |
|---|---|---|---|---|---|
| | Average ± SD | Average ± SD | Average ± SD | Average ± SD | Average ± SD |
| $PM_{2.5}$ ($\mu g/m^3$) | 106±89 | 98 ± 60 | 58±48 | 150±121 | 120±76 |
| OC ($\mu g/m^3$) | 16.0±15.8 | 8.5±5.2 | 10.3±7.4 | 28.9±22.0 | 14.6±10.8 |
| EC ($\mu g/m^3$) | 5.0±4.8 | 3.3±1.8 | 3.5±2.9 | 7.8±6.6 | 5.3±4.7 |
| OC/EC | 3.6±1.4 | 2.8±0.8 | 3.8±1.9 | 4.3±1.2 | 3.3±0.9 |
| WSOC ($\mu g/m^3$) | 6.5±6.5 | 4.4±3.6 | 5.2±4.0 | 10.3±9.8 | 5.9±4.9 |
| HULIS ($\mu g/m^3$) | 7.5±7.8 | 5.5±4.4 | 5.6±4.7 | 12.3±11.7 | 6.5±5.5 |
| $HULIS/PM_{2.5}$ (%) | 7.2±3.3 | 5.9±3.5 | 9.4±3.1 | 7.9±2.5 | 4.8±1.7 |
| $HULIS_C$/OC (%) | 24.5±8.3 | 29.2±6.2 | 26.2±9.6 | 21.0±7.1 | 22.0±6.9 |
| $HULIS_C$/WSOC (%) | 59.5±9.2 | 66.7±5.4 | 54.1±11.2 | 62.3±5.7 | 56.6±6.3 |
| $SO_4^{2-}$ ($\mu g/m^3$) | 22.3±24.1 | 22.6±17.0 | 10.9±13.2 | 32.7±35.1 | 22.5±16.5 |
| $NO_3^-$ ($\mu g/m^3$) | 18.6±18.0 | 17.2±13.4 | 10.8±13.2 | 20.1±17.8 | 29.0±23.8 |
| $Cl^-$ ($\mu g/m^3$) | 4.2±4.9 | 1.8±1.9 | 1.3±1.6 | 6.5±5.7 | 7.9±5.2 |
| $Na^+$ ($\mu g/m^3$) | 0.60±0.51 | 0.40±0.30 | 0.33±0.41 | 0.89±0.61 | 0.79±0.36 |
| $K^+$ ($\mu g/m^3$) | 2.2±2.6 | 2.2±2.9 | 1.3±1.0 | 3.2±3.6 | 2.2±1.3 |
| $Mg^{2+}$ ($\mu g/m^3$) | 0.18±0.19 | 0.15±0.07 | 0.18±0.08 | 0.24±0.32 | 0.10±0.07 |
| $Ca^{2+}$ ($\mu g/m^3$) | 0.97±0.57 | 0.99±0.52 | 1.14±0.48 | 0.83±0.70 | 0.89±0.46 |
| $NH_4^+$ ($\mu g/m^3$) | 14.1±13.0 | 13.2±9.8 | 6.6±7.0 | 19.1±16.9 | 18.4±11.8 |





**Table 2.** HULIS$_C$/OC and HULIS$_C$/WSOC values in the source samples

| Source type | Stove/vehicle | HULIS$_C$/OC | HULIS$_C$/WSOC | n |
|---|---|---|---|---|
| **Residential biofuel burning** | | | | |
| wood burning | improve stove | 0.41±0.07 | 0.62±0.06 | 3 |
| wheat straw | improve stove | 0.50±0.04 | 0.65±0.05 | 4 |
| corn stover | improve stove | 0.42±0.04 | 0.62±0.04 | 3 |
| **Residential chunk coal combustion** | | | | |
| SM, Var=32.4% | high efficiency heating stove | 0.14±0.07 | 0.51±0.04 | 3 |
| JY, Var=27.7% | high efficiency heating stove | 0.18±0.04 | 0.50±0.04 | 3 |
| BH, Var=25.0% | high efficiency heating stove | 0.08±0.02 | 0.44±0.01 | 3 |
| DT, Var=19.4% | high efficiency heating stove | 0.15 | 0.62 | 1 |
| SM, Var=32.4% | traditional cooking and heating stove | 0.06±0.01 | 0.46±0.02 | 3 |
| JY, Var=27.7% | traditional cooking and heating stove | 0.07±0.03 | 0.41±0.06 | 3 |
| BH, Var=25.0% | traditional cooking and heating stove | 0.05±0.01 | 0.43±0.08 | 3 |
| **Residential briquette coal combustion** | | | | |
| XM, Var=9.6% | high efficiency heating stove | 0.24±0.07 | 0.53±0.09 | 3 |
| **Vehicle exhaust** | | | | |
| traffic tunnel | mixed of gasoline and diesel vehicles | 0.05 | 0.65 | 1 |
| heavy-duty diesel trucks | Euro II | 0.16±0.02 | 0.38±0.03 | 3 |
| light-duty gasoline vehicles | Euro IV | 0.11±0.03 | 0.21±0.11 | 4 |







**Table 3.** Average and seasonal contributions percent of various sources to ambient HULIS concentrations in Beijing

622 (%)

|  | Residential biofuel burning | Residential coal burning | Transportation | Industries | Biomass open burning | Secondary process |
|---|---|---|---|---|---|---|
| Average | 57.4±9.1 | 12.3±2.8 | 1.5±0.3 | 1.1±0.3 | 1.7±0.5 | 25.9±9.3 |
| Summer | 36.3±8.7 | 7.8±2.6 | 2.9±0.9 | 2.4±1.2 | 10.3±3.5 | 40.2±18.1 |
| Autumn | 34.7±8.2 | 7.4±2.3 | 2.3±0.7 | 1.6±0.8 | 1.3±0.7 | 52.7±17.1 |
| Winter | 69.6±20.2 | 14.9±6.1 | 0.8±0.3 | 0.5±0.3 | 0.0±0.0 | 14.3±18.2 |
| Spring | 69.7±17.0 | 14.9±5.1 | 1.3±0.4 | 0.9±0.4 | 0.1±0.0 | 13.1±13.4 |

Note: only the sources with an average contribution over than 1% were provided. Uncertainty estimation for the
seasonal and annual primary and secondary HULISc contributions was determined using a bootstrap sampling
technique, which is described in Text S3.2. These uncertainties are based on the assumption that the uncertainty of the
$PPM_{2.5}$ and $f_{OC}$ values are 30% and 15%, respectively. Uncertainty calculations based on larger uncertainties (50% for
both $PPM_{2.5}$ and $f_{OC}$) show 5-10% higher relative uncertainties for the residential biofuel and secondary process but
small changes for other primary sectors (see Table S5).