# Peer review of "Quantifying primary and secondary humic-like substances in"

_Atmospheric Chemistry and Physics, 2018_

## Referee Comment (RC1) · Anonymous Referee #1 · 29 Oct 2018

The manuscript is a comprehensive and well-structured study on the potential sources of HULIS, a ubiquitous and abundant atmospheric aerosol constituent. Besides the fact that it is based on a surprisingly extensive experimental setup covering different source measurements and long-term field sampling and observations, it also has a touch of novelty in that identifies residential coal burning as a potentially important yet previously ignored source of primary HULIS. The methodology applied in the manuscript is widely accepted by the aerosol community and its use makes the comparison with the results of other publications feasible. Although the study involves only the analyses of key

aerosol constituents such as OC, WSOC, HULIS and inorganics, its conclusions are largely well-founded by the results of the measurements. There are just a few issues that raise some questions in the reviewer.

1) Except for the summer, HULIS are much better correlated with PM2.5 than with OC (Page 8, Line 220 and 224). This is surprising in the light of the fact that HULIS is actually part of OC whereas PM2.5 contains all sorts of other constituents. Not surprisingly, the correlations are the best for WSOC, the closest relative of HULIS. Is there any possible explanation for these observations? Perhaps the effects of vehicular exhaust contributing to OC (and EC) but less to PM2.5 mass concentrations?

2) In sub chapter 3.2 the differences in HULIS-to-OC ratios of biomass combustion emissions between this study and many other studies around the world are stunning. There are differences by factors of 3–5. The manuscript actually claims that nearly half of the OC are HULIS. Since these are emission measurements on biomass that should not be fundamentally different in different regions (albeit significant differences are seen between various species), there should be something in the experimental setup that causes these unusually high readings. Differences in combustion conditions, dryness of fuel, dilution ratios and excessive cooling may explain these high values. A comparative and critical assessment of the results with those of similar studies would be useful. This is critical since the source apportionment of primary HULIS is based on these emission values.

3) I would strongly discourage the application of simple correlations for secondary formation processes (sub chapter 3.4.2). These mechanisms are too complex to be captured by simple regressions: emission fluxes of precursors, rates of transformations, volatilities and water-solubilities of the reaction products, cloud-processing mechanisms, are all different and the processes are strongly non-linear. If, for example, HULIS is not correlated with sulfate, it may also mean that though they are both of secondary origin, the sources and emission fluxes of their precursors are very much different. Therefore lack of correlation does not indicate anything, neither does some
moderate virtual correlation. Just think of the examples of sulfate and nitrate, both being secondary aerosol constituents, yet they exhibit completely different formation mechanisms relative to the emissions of their precursors.

Minor comments:

Typography throughout the manuscript: the improper use of hyphen instead of En dash and Minus characters.

Page 5 Line 122 'systemis' . . . space missing

Page 5 Line 122 'induced' . . . introduced?

Page 5 Line 124 'at ambient temperature' . . .below ambient temperature?

Page 5 Line 142 'measurements was' . . .were

Page 6 Line 154 'determination' . . . determined

Page 7 Line 198 'General of ambient'

Page 9 Line 239 'HULIC'

Page 9 Line 245 Please define 'WSOM'

---

## Referee Comment (RC2) · Anonymous Referee #2 · 8 Dec 2018

This work integrates ambient, source sample measurements and modeling investigation to quantify HULIS sources in Beijing. This integrative approach provides quantitative insights into HULIS sources that otherwise are not easily extracted from source and ambient measurements alone. The paper is well–written and easy to follow. I have one main concern regarding the estimation of secondary HULIS. It is estimated to be the difference of measured HULIS and modelled primary HULIS. The difference method is inherently associated with large uncertainty and it appears less reliable (see more details in the specific comments). Any overestimate in primary HULIS would translate to

underestimate in secondary HULIS. It is desirable that the authors conduct a receptor model source apportionment (such as positive matrix factorization) using the measured chemical composition to estimate the secondary HULIS contribution and inter-compare with the results obtained from the CMAQ model.

Specific comments:

1. Model evaluation of HULIS. Fig. 4 compares predicted primary HULISc and observed HULISc on days with relative good primary PM2.5 model performance. In the main text it is reported fractional error of less than 0.6 was used to select the good model performance data. What is the percentage of data in this work's dataset fall outside this criterion of good modeling performance? Are there any patterns in the sub-group of data with poor agreement?

2. In this work, contributions of HULISc from secondary processes were determined by subtracting predicted primary HULISc from observed HULISc. The percentage contribution of secondary process was 40.2% in summer, 52.7% in fall, 14.3% in winter and 13.1% in spring. The secondary HULIS contribution was surprisingly low, considering the strong correlations of HULIS with secondary PM components such as sulfate and estimated SOC, especially for winter samples (Figure 3)

3. Related to the previous comment, and also the fact that on some days the predicted primary HULISc concentrations are greater than the observed HULISc, I have the concern whether certain assumptions made in the model have led to positive bias for primary HULISc (therefore negative bias for secondary HULISc) (e.g., assumption of foc values, see the next comment). How many samples were predicted by the model to have negative secondary HULISc? Are there any common characteristics in these samples that might shed some insights for the potential bias?

4. Table S3 lists the values of foc for primary sources considered in the model. "Residential" source has the largest foc at 62.80%. It appears this residential source is residential coal combustion (#91028) (Ying et al, 2018). Was this Residential source

foc also applied to residential biofuel burning? If yes, is there supporting evidence for this assumption? The apportionment of primary HULIS sources by the CMAQ model in this work suggested that residential biofuel burning was the largest HULIS source year around (34-70%), and especially dominant during winter and spring (70%). The foc in open biomass burning (arguably a burning activity bearing similarity to residential biofuel burning) is 29.40%, only $\sim$1/2 of the foc for residential coal combustion. Apparently, the foc value assumed has a large impact on the modeled source contribution. The authors need to clarify what foc value is adopted for residential biofuel combustion and the rationales behind.

5. Please comment on other potential primary HULIS source, such as cooking, which might make a contribution, but are not considered in the current model.

6. Table 2: provide a table footnote to briefly explain the abbreviations for the different residential coals.

7. Table S5: add a note to indicate the comparative relationship of this table with Table 3 in the main text.

---

## Author Comment (AC1) · 21 Dec 2018

The manuscript is a comprehensive and well-structured study on the potential sources of HULIS, a ubiquitous and abundant atmospheric aerosol constituent. Besides the fact that it is based on a surprisingly extensive experimental setup covering different source measurements and long-term field sampling and observations, it also has a touch of novelty in that identifies residential coal burning as a potentially important yet previously ignored source of primary HULIS. The methodology applied in the manuscript is widely accepted by the aerosol community and its use makes the comparison with the results of other publications feasible. Although the study involves only the analyses of key aerosol constituents such as OC, WSOC, HULIS and inorganics, its conclusions are largely well-founded by the results of the measurements. There are just a few issues that raise some questions in the reviewer.

**Response**: We thank the reviewer #1 for providing helpful comments and suggestions to improve our manuscript. Below are our responses to reviewer comments including descriptions how we have modified the manuscript.

**1)** Except for the summer, HULIS are much better correlated with PM2.5 than with OC (Page 8, Line 220 and 224). This is surprising in the light of the fact that HULIS is actually part of OC whereas PM2.5 contains all sorts of other constituents. Not surprisingly, the correlations are the best for WSOC, the closest relative of HULIS. Is there any possible explanation for these observations? Perhaps the effects of vehicular exhaust contributing to OC (and EC) but less to PM2.5 mass concentrations?

**Response**: Both HULIS and OC are strongly correlated with $PM_{2.5}$, indicating that they have similar sources such as biomass and coal burning, secondary processes. Perhaps the effects of vehicular exhaust contributing to OC (and EC) but less to PM2.5 mass concentrations, which need to explore in the future.

**2)** In sub chapter 3.2 the differences in HULIS-to-OC ratios of biomass combustion emissions between this study and many other studies around the world are stunning. There are differences by factors of 3–5. The manuscript actually claims that nearly half

of the OC are HULIS. Since these are emission measurements on biomass that should not be fundamentally different in different regions (albeit significant differences are seen between various species), there should be something in the experimental setup that causes these unusually high readings. Differences in combustion conditions, dryness of fuel, dilution ratios and excessive cooling may explain these high values. A comparative and critical assessment of the results with those of similar studies would be useful. This is critical since the source apportionment of primary HULIS is based on these emission values.

**Response**: We summarized the  HULISc/OC and HULIS/WSOC values from biomass burning (see the following Table). We think the combustion condition have much influence on the HULIS-to-OC ratios. For biomass open burning, HULIS-to-OC ratios varied less (from 0.14-0.35), while for biomass burned in the stove, ratios varied a lot (from 0.01-0.50). For advanced  stove used in European (with secondary air), combustion is relatively complete, thus HULIS produce less (0.01-0.12), while for stove used in Chinese rural household, combustion is relatively inadequate, thus HULIS produce more (0.41-0.50). Dilution ratio (DR) and residence time (RT) could affect gas-particle partitioning, and thus also have effect on the results (Lipsky et al., 2006; May et al., 2013). Dryness content of fuels was found to be not correlated with HULIS-to-OC ratios.

We added a comparative and critical assessment of the results with those of similar studies in the revised manuscript and the following table was added in the supplement.

Summary of HULISc/OC and HULIS/WSOC values from biomass burning

| Biomass | Combustion condition | Sampling condition | HULIS$_C$/OC | HULIS$_C$/WSOC | Reference |
|---|---|---|---|---|---|
| Wood (M=9.3%) | Improve stove | Chamber/hood DR≈40, RT≈80s | 0.41±0.07 | 0.62±0.06 | This study |
| Wheat straw (M=9.8%) | Improve stove | Chamber/hood DR≈40, RT≈80s | 0.50±0.04 | 0.65±0.05 | This study |
| Maize stover (M=8.0%) | Improve stove | Chamber/hood DR≈40, RT≈80s | 0.42±0.04 | 0.62±0.04 | This study |

| | | | | | |
|---|---|---|---|---|---|
| Wood (M=7~14.8%) | Chimney type logwood stove with primary/secondary air | Dilution source sampler with DR=10, RT long enough (no specified) | 0.04-0.11 | | Goncalves et al., 2010 |
| Wood (M=10~16%) | Domestic tile stove | Dilution sampler with DR=3, RT=0.2s | 0.01-0.12 | | Schmidl et al., 2008a |
| Leaves (M=25%) | Open burning | Smoke plume | 0.33-0.35 | | Schmidl et al., 2008b |
| Rice straw (M=5.8%) | Open burning | Chamber | | 0.66±0.02 | Fan et al., 2016 |
| Corn straw (M=7.4%) | Open burning | Chamber | | 0.59±0.02 | Fan et al., 2016 |
| Pine branch (M=7.6%) | Open burning | Chamber | | 0.57±0.03 | Fan et al., 2016 |
| Rice straw | Open burning and chamber | Chamber/hood or downwind | 0.34±0.05 | | Lin et al., 2010a |
| Sugarcane leaves | Open burning | Chamber/hood | 0.28±0.03 | | Lin et al., 2010a |
| Charcoal | Open burning | Downwind | 0.32 | | Lin et al., 2010a |
| Rice straw | Open burning | Downwind | 0.14 | 0.33±0.02 | Lin et al., 2010b |
| Sugarcane | Open burning | Downwind | 0.15 | 0.30±0.01 | Lin et al., 2010b |
| Rice straw (M=7.8%) | Open burning | Chamber/hood | 0.26±0.03 | 0.63±0.05 | Park and Yu, 2016 |
| Pine needles (M=9.9%) | Open burning | Chamber/hood | 0.15±0.04 | 0.36±0.08 | Park and Yu, 2016 |
| Sesame stems (M=10.3%) | Open burning | Chamber/hood | 0.29±0.08 | 0.51±0.08 | Park and Yu, 2016 |

Note: M, DR and RT are the abbreviations of Moisture, Dilution Ratio and Residence Time, respectively.

Reference:

Fan, X., Wei, S., Zhu, M., Song, J., and Peng, P.: Comprehensive characterization of humic-like substances in smoke $PM_{2.5}$ emitted from the combustion of biomass materials and fossil fuels, Atmos. Chem. Phys., 16, 13321–13340, 2016.

Goncalves, C., Alves, C., Evtyugina, M., Mirante, F., Pio, C., Caseiro, A., Schmidl, C., Bauer, H., and Carvalho F.: Characterisation of $PM_{10}$ emissions from woodstove combustion of common woods grown in Portugal, Atmos. Environ.,

44(35): 4474-4480, 2010.

Lin, P., Engling, G., and Yu, J.Z.: Humic-like substances in fresh emissions of rice straw burning and in ambient aerosols in the Pearl River Delta Region, China. Atmos. Chem. Phys., 10, 6487-6500, 2010a.

Lin, P., Huang, X.F., He, L.Y., and Yu, J.Z.: Abundance and size distribution of HULIS in ambient aerosols at a rural site in South China, J. Aerosol Sci., 41, 74–87, 2010b.

Park, S. S. and Yu, J.: Chemical and light absorption properties of humic-like substances from biomass burning emissions under controlled combustion experiments, Atmos. Environ., 136, 114-122, 2016.

Schmidl, C., Marr, L. L., Caseiro, A., Kotianova, P., Berner, A., Bauer, H., Kasper-Giebl, A., and Puxbaum, H. Chemical characterisation of fine particle emissions from wood stove combustion of common woods growing in mid-European Alpine regions, Atmos. Environ., 42, 126–141, 2008a.

Schmidl, C., Bauer, H., Dattler, A., Hitzenberger, R., Weissenboeck, G., Marr, I. L., and Puxbaum, H.: Chemical characterisation of particle emissions from burning leaves, Atmos. Environ., 42, 9070-9079, 2008b.

Lipsky, E. M., and Robinson, A. L.: Effects of dilution on fine particle mass and partitioning of semivolatile organics in diesel exhaust and wood smoke, Environ. Sci. Technol., 40(1), 155-162, 2006.

May, A. A., Levin, E. J. T., Hennigan, C. J., Riipinen, I., Lee, T., Collett, J. L., Jimenez, J. L., Kreidenweis, S. M., Robinson, A. L.: Gas-particle partitioning of primary organic aerosol emissions: 3. Biomass burning, Journal of Geophysical Research-Atmospheres, 118(19): 11327-11338, 2013.

**3)** I would strongly discourage the application of simple correlations for secondary formation processes (sub chapter 3.4.2). These mechanisms are too complex to be

captured by simple regressions: emission fluxes of precursors, rates of transformations, volatilities and water-solubilities of the reaction products, cloud-processing mechanisms, are all different and the processes are strongly non-linear. If, for example, HULIS is not correlated with sulfate, it may also mean that though they are both of secondary origin, the sources and emission fluxes of their precursors are very much different. Therefore lack of correlation does not indicate anything, neither does some moderate virtual correlation. Just think of the examples of sulfate and nitrate, both being secondary aerosol constituents, yet they exhibit completely different formation mechanisms relative to the emissions of their precursors.

**Response**: we have deleted the sub chapter 3.4.2.

**Minor comments:**

Typography throughout the manuscript: the improper use of hyphen instead of En dash and Minus characters.

Page 5 Line 122 'systemis' . . . space missing

Page 5 Line 122 'induced' . . . introduced?

Page 5 Line 124 'at ambient temperature' . . .below ambient temperature?

Page 5 Line 142 'measurements was' . . .were

Page 6 Line 154 'determination' . . . determined

Page 7 Line 198 'General of ambient'

Page 9 Line 239 'HULIC'

Page 9 Line 245 Please define 'WSOM'

**Response**: We revised these grammatical errors accordingly.

---

## Author Comment (AC2) · 21 Dec 2018

This work integrates ambient, source sample measurements and modeling investigation to quantify HULIS sources in Beijing. This integrative approach provides quantitative insights into HULIS sources that otherwise are not easily extracted from source and ambient measurements alone. The paper is well–written and easy to follow. I have one main concern regarding the estimation of secondary HULIS. It is estimated to be the difference of measured HULIS and modelled primary HULIS. The difference method is inherently associated with large uncertainty and it appears less reliable (see more details in the specific comments). Any overestimate in primary HULIS would translate to underestimate in secondary HULIS. It is desirable that the authors conduct a receptor model source apportionment (such as positive matrix factorization) using the measured chemical composition to estimate the secondary HULIS contribution and inter-compare with the results obtained from the CMAQ model.

**Response:** We thank the reviewer #2 for instructive comments to help us improve the manuscript. We have conducted a receptor model source apportionment (PMF) to estimate source contribution to ambient HULIS (including the secondary HULIS contribution) and inter-compared with the results obtained from the CMAQ model. The work has been submitted to "Science of Total Environment" for reviewing.

Below are our responses to reviewer comments including descriptions how we have modified the manuscript.

**Specific comments:**

**1.** Model evaluation of HULIS. Fig. 4 compares predicted primary HULISc and observed HULISc on days with relative good primary PM2.5 model performance. In the main text it is reported fractional error of less than 0.6 was used to select the good model performance data. What is the percentage of data in this work's dataset fall outside this criterion of good modeling performance? Are there any patterns in the sub-group of data with poor agreement?

**Response:** The percentage of data fall outside the "good" performance range in spring, summer, autumn and winter is approximately 12% (3/25), 30% (8/26), 55% (18/33) and 25% (7/27), respectively. We noticed that on these "bad" performance days, the model significantly overpredicted concentrations of $PPM_{2.5}$ in autumn and winter, with a mean fractional bias (MFB) of 1.16 and 0.64, respectively. For spring and summer, the model under-predicted $PPM_{2.5}$ with MFB of -0.39 and -0.21, respectively, on the bad performance days. In comparison, for good model performance days, the MFB values are -0.09 (spring), 0.15 (summer), -0.05 (autumn), and -0.08 (winter). The average concentrations of estimated $PPM_{2.5}$ during these bad performance days are 76 $\mu$g m$^{-3}$ (spring), 68 $\mu$g m$^{-3}$ (summer), 9 $\mu$g m$^{-3}$ (autumn) and 32 $\mu$g m$^{-3}$ (winter). In contrast, the averaged $PPM_{2.5}$ on the good performance days are 46 $\mu$g m$^{-3}$ (spring), 33 $\mu$g m$^{-3}$ (summer), 34 $\mu$g m$^{-3}$ (autumn) and 103 $\mu$g m$^{-3}$ (winter). From this analysis, it is evident that the observed $PPM_{2.5}$ concentrations on the bad model performance days are quite different from that on the good performance days. The CMAQ model performance decreases when the observed concentrations are higher or lower than the normal concentration for that season. It is probably because that the day-to-day variations in the emission are poorly represented in the emission processing (currently, only weekday-weekend differences are considered in each month). The good day results reported in this study are representative of common conditions within each season.

**2.** In this work, contributions of HULISc from secondary processes were determined by subtracting predicted primary HULISc from observed HULISc. The percentage contribution of secondary process was 40.2% in summer, 52.7% in fall, 14.3% in winter and 13.1% in spring. The secondary HULIS contribution was surprisingly low, considering the strong correlations of

HULIS with secondary PM components such as sulfate and estimated SOC, especially for winter samples (Figure 3).

**Response:** According your Comment 4, we double checked foc and HULISc/OC data sources and revised the data (see detail in Comment 4 Response). The revised calculation leads to more secondary HULISc. The percentage contribution of secondary process is 50.2% in summer, 63.2% in fall, 30.3% in winter and 25.4% in spring, with annual average contribution of 38.9%.

**3.** Related to the previous comment, and also the fact that on some days the predicted primary HULISc concentrations are greater than the observed HULISc, I have the concern whether certain assumptions made in the model have led to positive bias for primary HULISc (therefore negative bias for secondary HULISc) (e.g., assumption of foc values, see the next comment). How many samples were predicted by the model to have negative secondary HULISc? Are there any common characteristics in these samples that might shed some insights for the potential bias?

**Response:** Thirty-two (32) days out of 72 have negative secondary HULISc. However, the negative values are usually very low (-1.40±1.49, one standard deviation). These days are usually associated with low total HULISc concentrations (2.73±3.10). Thus, this treatment does not introduce significant bias in the estimation of secondary HULISc overall.

**4.** Table S3 lists the values of foc for primary sources considered in the model. "Residential" source has the largest foc at 62.80%. It appears this residential source is residential coal combustion (#91028) (Ying et al, 2018). Was this Residential source foc also applied to residential biofuel burning? If yes, is there supporting evidence for this assumption? The apportionment of primary HULIS sources by the CMAQ model in this work suggested that residential biofuel burning was the largest HULIS source year around (34-70%), and especially dominant during winter and spring (70%). The foc in open biomass burning (arguably a burning activity bearing similarity to residential biofuel burning) is 29.40%, only _1/2 of the foc for residential coal combustion. Apparently, the foc value assumed has a large impact on the modeled source contribution. The authors need to clarify what foc value is adopted for residential biofuel combustion and the rationales behind.

**Response:** Thanks for the suggestion. #91028 is for residential coal combustion. We have adopted foc for residential biofuel burning at 42.51% based on field measurement in China (Li et al., 2009).

*Li X., Wang S., Duan L., Hao J., Nie Y. Carbonaceous aerosol emissions from household biofuel combustion in China. Environmental Science & Technology, 2009, 43: 6076-6081.*

We double checked foc data source of open burning and found that #92084 is not for biomass open burning, thus we removed it and averaged #92000 and #92090 and obtained the foc for open biomass burning at 42.29%.

We adopted HULISc/OC for open burning same as biofuel combustion (44%, from our measurement) previously. However, when we compared HULISc/OC from open burning and biofuel combustion, we found difference between them. For biomass open burning, HULIS-to-OC ratios varied less (from 0.14-0.35), while for biomass burned in the stove, ratios varied a lot (from 0.01-0.50). For advanced stove used in European (with secondary air), combustion is relatively complete, thus HULIS produce less (0.01-0.12), while for stove used in Chinese rural household, combustion is relatively inadequate, thus HULIS produce more (0.41-0.50). We think combustion condition has much influence on the HULIS-to-OC ratios. For stove used in European and China, biofuel is burned in a relatively enclosed combustion chamber. Dilution ratio (DR) and residence time (RT) could affect gas-particle partitioning, and thus also have effect on the results (Lipsky et al., 2006; May et al., 2013). Thus we adopted HULISc/OC for open burning at 25% (average value of previous references about biomass open burning).

| Biomass | Combustion condition | Sampling condition | HULIS$_C$/OC | HULIS$_C$/WSOC | Reference |
|---------|---------------------|--------------------|--------------|----------------|-----------|
| Wood (M=9.3%) | Improve stove | Chamber/hood DR≈40, RT≈80s | 0.41±0.07 | 0.62±0.06 | This study |
| Wheat straw (M=9.8%) | Improve stove | Chamber/hood DR≈40, RT≈80s | 0.50±0.04 | 0.65±0.05 | This study |
| Maize stover (M=8.0%) | Improve stove | Chamber/hood DR≈40, RT≈80s | 0.42±0.04 | 0.62±0.04 | This study |
| Wood | Chimney type | Dilution source sampler with DR=10, RT long enough (no | 0.04-0.11 | | Goncalves et al., 2010 |

| Material | Stove/Burning | Sampling | | | Reference |
|---|---|---|---|---|---|
| (M=7~14.8%) | logwood stove with primary/secondary air | specified) | | | |
| Wood (M=10~16%) | Domestic tile stove | Dilution sampler with DR=3, RT=0.2s | 0.01-0.12 | | Schmidl et al., 2008a |
| Leaves (M=25%) | Open burning | Smoke plume | 0.33-0.35 | | Schmidl et al., 2008b |
| Rice straw (M=5.8%) | Open burning | Chamber | | 0.66±0.02 | Fan et al., 2016 |
| Corn straw (M=7.4%) | Open burning | Chamber | | 0.59±0.02 | Fan et al., 2016 |
| Pine branch (M=7.6%) | Open burning | Chamber | | 0.57±0.03 | Fan et al., 2016 |
| Rice straw | Open burning and chamber | Chamber/hood or downwind | 0.34±0.05 | | Lin et al., 2010a |
| Sugarcane leaves | Open burning | Chamber/hood | 0.28±0.03 | | Lin et al., 2010a |
| Charcoal | Open burning | Downwind | 0.32 | | Lin et al., 2010a |
| Rice straw | Open burning | Downwind | 0.14 | 0.33±0.02 | Lin et al., 2010b |
| Sugarcane | Open burning | Downwind | 0.15 | 0.30±0.01 | Lin et al., 2010b |
| Rice straw (M=7.8%) | Open burning | Chamber/hood | 0.26±0.03 | 0.63±0.05 | Park and Yu, 2016 |
| Pine needles (M=9.9%) | Open burning | Chamber/hood | 0.15±0.04 | 0.36±0.08 | Park and Yu, 2016 |
| Sesame stems (M=10.3%) | Open burning | Chamber/hood | 0.29±0.08 | 0.51±0.08 | Park and Yu, 2016 |

Note: M, DR and RT are the abbreviations of Moisture, Dilution Ratio and Residence Time, respectively.

Reference:

*Fan, X., Wei, S., Zhu, M., Song, J., and Peng, P.: Comprehensive characterization of humic-like substances in smoke $PM_{2.5}$ emitted from the combustion of biomass materials and fossil fuels, Atmos. Chem. Phys., 16, 13321–13340, 2016.*

*Goncalves, C., Alves, C., Evtyugina, M., Mirante, F., Pio, C., Caseiro, A., Schmidl, C., Bauer, H., and Carvalho F.: Characterisation of $PM_{10}$ emissions from woodstove combustion of common woods grown in Portugal, Atmos. Environ., 44(35): 4474-4480, 2010.*

*Lin, P., Engling, G., and Yu, J.Z.: Humic-like substances in fresh emissions of rice straw burning and in ambient aerosols in the Pearl River Delta Region, China. Atmos. Chem. Phys., 10, 6487-6500, 2010a.*

*Lin, P., Huang, X.F., He, L.Y., and Yu, J.Z.: Abundance and size distribution of HULIS in ambient aerosols at a rural site in South China, J. Aerosol Sci., 41, 74–87, 2010b.*

*Park, S. S. and Yu, J.: Chemical and light absorption properties of humic-like substances from biomass burning emissions under controlled combustion experiments, Atmos. Environ., 136, 114-122, 2016.*

*Schmidl, C., Marr, L. L., Caseiro, A., Kotianova, P., Berner, A., Bauer, H., Kasper-Giebl, A., and Puxbaum, H. Chemical characterisation of fine particle emissions from wood stove combustion of common woods growing in mid-European Alpine regions, Atmos. Environ., 42, 126–141, 2008a.*

*Schmidl, C., Bauer, H., Dattler, A., Hitzenberger, R., Weissenboeck, G., Marr, I. L., and Puxbaum, H.: Chemical characterisation of particle emissions from burning leaves, Atmos. Environ., 42, 9070-9079, 2008b.*

*Lipsky, E. M., and Robinson, A. L.: Effects of dilution on fine particle mass and partitioning of semivolatile organics in diesel exhaust and wood smoke, Environ. Sci. Technol., 40(1), 155-162, 2006.*

*May, A. A., Levin, E. J. T., Hennigan, C. J., Riipinen, I., Lee, T., Collett, J. L., Jimenez, J. L., Kreidenweis, S. M., Robinson, A. L.: Gas-particle partitioning of primary organic aerosol emissions: 3. Biomass burning, Journal of Geophysical Research-Atmospheres, 118(19): 11327-11338, 2013.*

**5.** Please comment on other potential primary HULIS source, such as cooking, which might make a contribution, but are not considered in the current model.

**Response:** We have mentioned other potential primary HULIS source, such as terrestrial and marine emissions, which were not included in these estimations of primary HULIS emissions since they were considered to be negligible for inland cities, such as Beijing (Graber and Rudich, 2006; Zheng et al., 2013).

Cooking contribute about twenty percent of ambient fine organic aerosols in Beijing (Wang et al., 2009; Zhang et al., 2016; Sun et al., 2016). Since cooking emissions was not included in MEIC, and no HULIS emission information about cooking is available, thus cooking are not considered in the current model. It might make a contribution to ambient HULIS and need to be explored in the future.

*Graber, E.R. and Rudich, Y.: Atmospheric HULIS: How humic-like are they? A comprehensive and critical review, Atmos. Chem. Phys., 6, 729-753, 2006.*

*Zheng, G. J., He, K.B., Duan, F.K., Cheng, Y., and Ma, Y. L.: Measurement of humic-like substances in aerosols: A review, Environ. Pollut., 181, 301-314, 2013.*

*Wang, Q., Shao, M., Zhang, Y., Wei, Y., Hu, M., and Guo, S.: Source apportionment of fine organic aerosols in Beijing. Atmos. Chem. Phys., 9, 8573–8585, 2009.*

*Zhang, Y. M., Wang, Y. Q., Zhang, X. Y., et al.: Chemical components, variation, and source identification of PM1 during the heavy air pollution episodes in Beijing in December 2016. J. Meteor. Res., 32(1), 1–13, doi: 10.1007/s13351-018-7051-8, 2018.*

*Sun, Y., Du, W., Fu, P., Wang, Q., Li, J., Ge, X., Zhang, Q., Zhu, C., Ren, L., and Xu, W.: Primary and secondary aerosols in Beijing in winter: sources, variations and processes, Atmos. Chem. Phys., 16 (13), 8309−8329, 2016.*

**6**. Table 2: provide a table footnote to briefly explain the abbreviations for the different residential coals.

**Response:** we have added a table footnote to briefly explain the abbreviations for the different residential coals.

Note: SM, DT indicate that coals come from the coal mines in ShenMu of Shaanxi Province and DaTong of Shanxi Province in China, respectively. JY and BH were supplied by two companies with the name of JiuYang and BeiHua, respectively, and no producing area of coal were not available. XM indicates briquette coal, which is the abbreviation of briquette coal in Chinese (XingMei).

**7**. Table S5: add a note to indicate the comparative relationship of this table with Table 3 in the main text.

**Response:** we have added a note to indicate the comparative relationship of this table with Table 3 in the main text.

Note: only the sources with an average contribution over than 1% were provided. Uncertainty estimation for the seasonal and annual primary and secondary HULISc contributions was determined using a bootstrap sampling technique, which is described in Text S3.2. These uncertainties are based on the assumption that the uncertainty for both $PPM_{2.5}$ and $f_{OC}$ values are 50%. Uncertainty calculations based on less uncertainties (30% for $PPM_{2.5}$ and 15% for $f_{OC}$) are shown in Table 3 in main text.

---

## Editor Decision (ED1)

I have the following two remarks on the content:

| Line | Remark |
|------|--------|
| 292 | POC and secondary organic carbon (SOC) were estimated using the EC tracer method (Lim and Turpin, 2002; Turpin and Huntzicker, 1995). : This statement might deserve a comment that comparison with AMS results has shown that this method generally overestimated the POC, and thus underestimates SOC. Also, one could mention " POC was calculated to be 2xEC (this is what I read from Fig. 2). |
| 294 | Figures 2a and 2b show that there are strong correlations between HULIS and POC, and HULIS and EC: This deserves a statement that it is not surprising that HULIS correlates with POC if it does so with EC, as POC is calculated from EC. |

In addition, the manuscript requires technical corrections. Please improve the wording in the following instances (incomplete list):

| Line | Text |
|------|------|
| 20 | Average concentration of ambient HULIS was 7.5 μg/m3 in atmospheric PM2.5 |
| 24 | shows residential biofuel and coal burning, secondary formation are important annual sources |
| 137 | to perform on-road emission test |
| 140 | draw a constant ratio of sample flow from exhaust |
| 152 | is provided in Text S1 of Supplement |
| 180 | PPM2.5,i is the calculated source contributions |
| 195 | Table S4 of Supplement |
| 198 | General of ambient aerosol characteristics |
| 209 | it is higher measurements in the urban areas |
| 216 | were similar with those |
| 219 | summarized in Table S1 of Supplement. |
| 220 | had a significant positive correlation with the annual r2=0.90 |
| 223 | lower than the ~10% in the PRD region |
| 224 | Strong correlations of HULISC with OC were observed with the annual r2=0.87 (and further instances of the same type: at least add a comma, otherwise it is confusing) |
| 236 | listed in Table S1 of Supplement (and further instances) |
| 251 | Combustion condition have much influence |
| 253 | For advanced stove used in |
| 254 | thus HULIS produce less; and further instances |
| 254 | While for stove used in Chinese rural household |
| 256 | and thus also have effect on the results |
| 257 | Dryness content of fuels was found to be |
| 269 | (MEP of China, 2014), |
| 272 | Due to lack of $f_{HULIS,i}$ |
| 279 | While industry sector and power plants contribute about 3% and close to zero, respectively. |
| 283 | Cooking contribute about twenty percent |
| 286 | thus cooking are not considered |
| 299 | biomass burning, industry, and vehicles contributes the rest. |
| 301 | $K_+$ generally originate from biomass burning with lesser contributions from coal burning and dust |
| 319 | about 200 Km |
| 321 | While weaker correlations were observed in summer and autumn with r2=0.40 and r2=0.43, respectively. |
| 326 | ($R^2$=0.89): otherwise always used $r^2$. |
| 328 | Significant correlation between …. were also found |
| 341 | were much higher than predicted primary HULISc concentrations |
| 353 | This difference is likely with the result of greater biofuel burning during the heating seasons in the Beijing area |
| 355 | A large contribution from residential sector |

366       Contributions from secondary processes also show obvious seasonal variations trend.

377       Figure 4 shows scatter plot

379       The variation of correlation coefficient

392       Appel et al.: replace by final version

603       represented by different shaped points

612       Concentrations of each seasons

617       percentages of several species to some others: e.g. WSOC is not a species

621       Table 2: improve stove
              mixed of gasoline and diesel vehicles

631       average contribution over than 1%

In addition:

Symbols should be italic.

References need harmonization in style.; e.g., paper titles should not be capital

SI

This needs editing as well.

---

## Author Response (AR2)

RESPONSES TO EDITORIAL COMMENTS

I have the following two remarks on the content:

Line Remark

POC and secondary organic carbon (SOC) were estimated using the EC tracer method (Lim and Turpin, 2002; Turpin and Huntzicker, 1995). : This statement might deserve a comment that comparison with AMS results has shown that this method generally overestimated the POC, and thus underestimates SOC. Also, one could mention " POC was calculated to be 2xEC (this is what I read from Fig. 2).

Response: We do not think it is appropriate to add such as statement since the AMS analyses are also subject to substantial uncertainties. Their estimation of POC is dependent on the amount of a particular fragment (m/e=44). Given the high energy electron impact ionization (70 eV), there could be alkane-like fragments broken from SOA molecules. There is not yet a clear way to really calibrate either method to assure the accuracy of the separation of POC and SOC and thus, it is really not useful to add this statement in this paper.

Figures 2a and 2b show that there are strong correlations between HULIS and POC, and HULIS and EC: This deserves a statement that it is not surprising that HULIS correlates with POC if it does so with EC, as POC is calculated from EC.

Response: Yes, if HULIS correlates with EC, then it should also correlate with POC. We have added text to this effect.

In addition, the manuscript requires technical corrections. Please improve the wording in the following instances (incomplete list):

Line Text

Average concentration of ambient HULIS was 7.5 μg/m3 in atmospheric PM2.5

Response: It was revised to "Average concentration of ambient HULIS in $PM_{2.5}$ was 7.5 μg/m$^3$".

shows residential biofuel and coal burning, secondary formation are important annual sources

Response: It was revised to "shows residential biofuel and coal burning, secondary formation are important sources".

to perform on-road emission test

Response: It was revised to "to perform on-road emission tests".

draw a constant ratio of sample flow from exhaust

Response: It was revised to "draw a constant ratio of sample flow to exhaust flow".

is provided in Text S1 of Supplement

Response: It was revised to "is provided in Text S1 of the Supplement".

PPM2.5,i is the calculated source contributions

Response: It was revised to "PPM2.5,i is the calculated source contribution".

Table S4 of Supplement

Response: It was revised to "Table S4 of the Supplement".

General of ambient aerosol characteristics

Response: It was revised to "General characteristics of ambient aerosol".

it is higher measurements in the urban areas

Response: It was revised to "it is higher than those measurements in the urban areas".

were similar with those

Response: It was revised to "were similar to those".

summarized in Table S1 of Supplement.

Response: It was revised to "summarized in Table S1 of the Supplement".

had a significant positive correlation with the annual r2=0.90

Response: It was revised to "had a significant correlation with the annual r2=0.90".

lower than the ~10% in the PRD region

Response: It was revised to " and was approximately 10% lower than that in in the PRD region".

Strong correlations of HULISC with OC were observed with the annual r2=0.87 (and
       further instances of the same type: at least add a comma, otherwise it is confusing)

Response: a comma was added in these sentence and further instances of the same type listed in Table S1 of Supplement (and further instances)

Response: It was revised to "listed in Table S1 of the Supplement". And further instances were
also revised.

Combustion condition have much influence

Response:    It was revised to "Combustion conditions have much influence".

For advanced stove used in

Response: It was revised to "For those advanced stoves used in"

thus HULIS produce less; and further instances

Response: It was revised to "thus HULIS was generated less", line 255, "thus HULIS produce
more" was also revised to "thus HULIS was generated more".

While for stove used in Chinese rural household

Response: It was revised to "While for the stoves used in Chinese rural households"

and thus also have effect on the results

Response: It was revised to "and thus have effect on the HULIS-to-OC ratios".

Dryness content of fuels was found to be

Response: It was revised to "Moisture content of fuels was found to be".

(MEP of China, 2014),

Response: It was revised to "(Ministry of Environment Protection of China, 2014)".

Due to lack of fHULIS,i

Response: It was revised to "Due to lack of the information of fHULIS,i ".

While industry sector and power plants contribute about 3% and close to zero,
       respectively.

Response: It was revised to "While industry sector and power plants contribute about 3% and
       close to zero of the annual primary HULIS emissions, respectively".

Cooking contribute about twenty percent

Response: It was revised to "Cooking contributes about twenty percent".

thus cooking are not considered

Response: It was revised to "thus cooking is not considered".

biomass burning, industry, and vehicles contributes the rest.

Response: It was revised to "biomass burning, industry, and vehicles contribute the rest".

K+ generally originate from biomass burning with lesser contributions from coal burning and dust

Response: It was revised to "K+ generally originates from biomass burning with lesser contributions from coal burning and dust".

about 200 Km

Response: It was revised to "about 200 kilometres".

While weaker correlations were observed in summer and autumn with r2=0.40 and r2=0.43, respectively.

Response: It was revised to "While weaker correlations between HULIS and Cl⁻ were observed in summer and autumn with $r^2$=0.40 and $r^2$=0.43, respectively".

(R2=0.89): otherwise always used r2.

Response: R2 was revised to r2

Significant correlation between …. were also found

Response: It was revised to "Significant correlations between …. were also found".

were much higher than predicted primary HULISc concentrations

Response: It was revised to "were much higher than the predicted primary HULISc concentrations".

This difference is likely with the result of greater biofuel burning during the heating seasons in the Beijing area

Response: It was revised to "This difference is likely with greater biofuel burning during the heating seasons in the Beijing area".

A large contribution from residential sector

Response: It was revised to "Great contribution from residential sector".

Contributions from secondary processes also show obvious seasonal variations trend.

Response:   It was revised to "Contribution from secondary processes also shows obvious seasonal variations trend".

Figure 4 shows scatter plot

Response: It was revised to "Figure 4 shows the scatter plot".

The variation of correlation coefficient

Response: It was revised to "The variation of correlations".

Appel et al.: replace by final version

Response: It was replace by final version.

represented by different shaped points

Response: It was revised to "represented by different shapes".

Concentrations of each seasons

Response: It was revised to "Concentrations of different seasons".

percentages of several species to some others: e.g. WSOC is not a species

Response: species was revised to compounds

Table 2: improve stove     mixed of gasoline and diesel vehicles

Response: improve stove was revised to improved stove, mixed was revised to mixture average contribution over than 1%

Response: It was revised to "average contribution over 1%".

In addition:

Symbols should be italic. References need harmonization in style.; e.g., paper titles should not be capital

Response: The references have been checked and modified as needed. However, there are some proper names that require capitalization.

SI

This needs editing as well.

Response: The SI has been reviewed and modified as needed.

[revised manuscript text omitted]

[a] Analysis by CHNS elemental analyzer (Vario EL, Elementar, Langenselbold, Germany)

**Table S3.** Summary of HULISc/OC and HULIS/WSOC values from biomass burning

| Biomass | Combustion condition | Sampling condition | HULIS$_C$/OC | HULIS$_C$/WSOC | Reference |
|---|---|---|---|---|---|
| Wood (M=9.3%) | Improve stove | Chamber/hood DR≈40, RT≈80s | 0.41±0.07 | 0.62±0.06 | This study |
| Wheat straw (M=9.8%) | Improve stove | Chamber/hood DR≈40, RT≈80s | 0.50±0.04 | 0.65±0.05 | This study |
| Maize stover (M=8.0%) | Improve stove | Chamber/hood DR≈40, RT≈80s | 0.42±0.04 | 0.62±0.04 | This study |
| Wood (M=7~14.8%) | Chimney type logwood stove with primary/secondary air | Dilution source sampler with DR=10, RT long enough (no specified) | 0.04-0.11 | | Goncalves et al., 2010 |
| Wood (M=10~16%) | Domestic tile stove | Dilution sampler with DR=3, RT=0.2s | 0.01-0.12 | | Schmidl et al., 2008a |
| Leaves (M=25%) | Open burning | Smoke plume | 0.33-0.35 | | Schmidl et al., 2008b |
| Rice straw (M=5.8%) | Open burning | Chamber | | 0.66±0.02 | Fan et al., 2016 |
| Corn straw (M=7.4%) | Open burning | Chamber | | 0.59±0.02 | Fan et al., 2016 |
| Pine branch (M=7.6%) | Open burning | Chamber | | 0.57±0.03 | Fan et al., 2016 |
| Rice straw | Open burning and chamber | Chamber/hood or downwind | 0.34±0.05 | | Lin et al., 2010a |
| Sugarcane leaves | Open burning | Chamber/hood | 0.28±0.03 | | Lin et al., 2010a |
| Charcoal | Open burning | Downwind | 0.32 | | Lin et al., 2010a |
| Rice straw | Open burning | Downwind | 0.14 | 0.33±0.02 | Lin et al., 2010b |
| Sugarcane | Open burning | Downwind | 0.15 | 0.30±0.01 | Lin et al., 2010b |
| Rice straw (M=7.8%) | Open burning | Chamber/hood | 0.26±0.03 | 0.63±0.05 | Park and Yu, 2016 |
| Pine needles (M=9.9%) | Open burning | Chamber/hood | 0.15±0.04 | 0.36±0.08 | Park and Yu, 2016 |
| Sesame stems (M=10.3%) | Open burning | Chamber/hood | 0.29±0.08 | 0.51±0.08 | Park and Yu, 2016 |

Note: M, DR and RT are the abbreviations of Moisture, Dilution Ratio and Residence Time, respectively.

**Table S4**. Values of $f_{OC}$ used in this study.

| Source | $f_{OC}$ | data source |
|---|---|---|
| Dust | 0.69% | 413502.5[a] |
| Residential coal combustion | 62.80% | 91028[a] |
| Residential biofuel burning | 42.51% | Li et al., 2009 |
| Transportation | 51.17% | 90% 91022 + 10% 3914[a] |
| Power | 2.63% | 91104[a] |
| Industry | 8.00% | 900162.5[a] |
| open burning | 29.40% | average of 92000, 92090[a] |

Note: US EPA SPECIATE database profile #

**Table S5.** Annual and seasonal contributions percent of anthropogenic various primary emission of HULIS in Beijing (%)

| Source types | Annual | Spring | Summer | Autumn | Winter |
|---|---|---|---|---|---|
| Power plants | 0.0 | 0.0 | 0.0 | 0.0 | 0.0 |
| Industries | 2.9 | 4.8 | 6.2 | 3.3 | 1.4 |
| Residential coal burning | 24.6 | 23.8 | 23.2 | 24.4 | 25.2 |
| Residential biofuel burning | 70.8 | 68.6 | 66.9 | 70.3 | 72.5 |
| Transportation | 1.7 | 2.8 | 3.7 | 2.0 | 0.8 |

**Table S6.** Average and seasonal contributions percent of various sources to ambient HULIS concentrations in Beijing (%) using relative uncertainties of 50% for both $PPM_{2.5}$ and $f_{OC}$.

| | Residential biofuel burning | Residential coal burning | Transportation | Industries | Biomass open burning | Secondary process |
|---|---|---|---|---|---|---|
| Annual | 46.9±9.5 | 15.1±3.7 | 2±0.4 | 1.3±0.3 | 1.7±0.6 | 39.1±12 |
| Summer | 29.1±9 | 9.4±3.4 | 3.9±1.4 | 2.9±1.4 | 10.3±4.7 | 50.3±20.6 |
| Autumn | 24.7±7.4 | 7.9±2.8 | 2.7±1 | 1.7±0.9 | 1.1±0.7 | 63.2±19.3 |
| Winter | 55.7±20.8 | 17.9±8.1 | 1.1±0.5 | 0.6±0.3 | 0±0 | 30.6±24.1 |
| Spring | 62.2±17.7 | 20.1±6.7 | 2±0.6 | 1.2±0.5 | 0.1±0.1 | 25.5±18.5 |

Note: only the sources with an average contribution over than 1% were provided. Uncertainty estimation for the seasonal and annual primary and secondary HULISc contributions was determined using a bootstrap sampling technique, which is described in Text S3.2. These uncertainties are based on the assumption that the uncertainty for both $PPM_{2.5}$ and $f_{OC}$ values are 50%. Uncertainty calculations based on less uncertainties (30% for $PPM_{2.5}$ and 15% for $f_{OC}$) are shown in Table 3 in main text.

**Figures**

[revised manuscript text omitted]